
# Local and synoptic meteorological influences on daily variability of summertime surface ozone in eastern China

Han Han[1], Jane Liu[1,2], Lei Shu[1], Tijian Wang[1], Huiling Yuan[1]

[1]School of Atmospheric Sciences, Nanjing University, Nanjing, China

[2]Department of Geography and Planning, University of Toronto, Toronto, Canada

Correspondence: Jane Liu (janejj.liu@utoronto.ca)

## Abstract

Ozone pollution in China is influenced by meteorological processes on multiple scales. Using multiple linear regression and weather classification, we statistically assess the impacts of local and synoptic meteorology on daily variability of surface ozone in eastern China in summer during 2013-2018. In this period, summertime surface ozone in eastern China (110-130ºE, 20-42ºN) is among the highest in the world with regional means of 73.1 and 114.7 μg m$^{-3}$, respectively, in daily mean and daily maximum 8-hour average. By developing a multiple linear regression (MLR) model driven by local and synoptic weather factors, we establish a quantitative linkage between the daily ozone concentrations and meteorology in the study region. The meteorology described by the MLR model can explain ~46% of the daily variability in summertime surface ozone across eastern China. The model shows that synoptic factors contribute to ~37% of the overall meteorological effects on daily variability of surface ozone in eastern China. Among local meteorological factors, relative humidity is the most influential variable in the center and south of eastern China including the Yangtze River Delta and the Pearl River Delta regions, while temperature is the most influential variable in the north covering the Beijing-Tianjin-





Hebei region. To further examine the synoptic influence of weather conditions explicitly, six

predominant synoptic weather patterns (SWPs) over eastern China in summer are objectively identified

using the self-organizing map clustering technique. The six SWPs are formed under the integral

influence of the East Asian summer monsoon, the western Pacific subtropical high, the Meiyu front, and

the typhoon activities. The results show that the impacts of each of the SWPs on the daily variability of

surface ozone vary largely inside the study area. The maximum impact can reach ±8 μg m$^{-3}$ or ±16% of

the daily mean over some subregions in eastern China. A combination of the regression and the

clustering approaches suggests a strong performance of the MLR model in predicting the sensitivity of

surface ozone in eastern China to the variation of synoptic weather. Our assessment highlights the

important role of meteorology in modulating ozone pollution over China.

**1 Introduction**

Surface ozone is a major air pollutant detrimental to human health and vegetation growth (Yue et al.,

2017). Ozone exposures are estimated to be associated with near 0.3 million premature deaths globally

in one year (Cohen et al., 2017; Liang et al., 2018). The dominant source of surface ozone is the

photochemical oxidation of volatile organic compounds (VOCs) and carbon monoxide (CO) in the

presence of nitrogen oxides (NO$_x$) (Monks et al., 2015). In the past decades, China has been suffering

from severe ozone pollution, causing a worldwide concern (Verstraeten et al., 2015). High ozone

concentrations exceeding China national air quality standard (200 and 160 μg m$^{-3}$, respectively, for

hourly and 8-hourly maximum values) occur frequently in major Chinese cities in the three most

developed regions, the Beijing-Tianjin-Hebei (BTH) region (T. Wang et al., 2006a; G. Li et al., 2017),

the Yangtze River Delta (YRD) (Shu et al., 2016, 2019), and the Pearl River Delta (PRD) (Y. Wang et

al., 2017; H. Wang et al., 2018). An increasing trend of 1-3% per year in surface ozone since 2000 is

observed at urban and regional background sites in the three city clusters (Y. Wang et al., 2012; Zhang



et al., 2014; Ma et al., 2016; Sun et al., 2016; Gao et al., 2017) and at a global baseline station in western China (Xu et al., 2016).

Surface ozone concentrations in China largely depend on emissions and meteorology (Han et al., 2018a, 2019). Anthropogenic and natural emissions from both native and foreign sources provide precursors for the formation of high ozone levels in China (Han et al., 2019), while meteorology can influence surface ozone variations from instantaneous to decadal scale through its modulation of various chemical and physical processes (T. Wang et al., 2017). On a decadal scale, both observations (Zhou et al., 2013) and simulations (S. Li et al., 2018) show surface ozone in southern China correlates positively to the strength of the East Asian summer monsoon (EASM).

The daily variation of surface ozone in China is sensitive to synoptic weather systems, as illustrated by studies for BTH (Zhang et al., 2012; Huang et al., 2015), YRD (Shu et al., 2016, 2019), PRD (Zhang et al., 2013; Jiang et al., 2015), and other regions of China (Tan et al., 2018). Frontal systems can drive the transboundary transport of ozone in northern China (Ding et al., 2015; Dufour et al., 2016). Downdrafts in the periphery circulation of a typhoon system can strongly enhance surface ozone before the typhoon landing in eastern or southern China (Jiang et al., 2015; Shu et al., 2016). Zhao and Wang (2017) suggested that a stronger western Pacific subtropical high (WPSH) can lead to lower surface ozone concentration over southern China and higher one over northern China in summer. Moreover, surface ozone concentrations also vary with mesoscale weather systems in hours (Hu et al., 2018), such as the mountain-valley circulation (T. Wang et al., 2006b) and the land-sea breezes (H. Wang et al., 2018). Despite the valuable mechanisms of how weather systems mentioned above influences ozone concentrations in China reported by previous studies, quantified ozone anomalies resulted from these systems are lacked.





Weather systems at different scales bring about changes in meteorological variables and thus surface ozone through their impacts on chemistry and physical processes. However, the relative importance of

various meteorological factors to surface ozone concentrations in China are still unclear. Previous studies suggested the importance of temperature, relative humidity, and winds to surface ozone in different regions (Lou et al., 2015; Pu et al., 2017; Zhan et al., 2018). The key influential meteorological factors vary from cities to cities in China (Gong et al., 2018; Chen et al., 2019). In general, high ozone episodes commonly appear under weak wind, high temperature, low humidity, and clear conditions.

These weather conditions can enhance stagnation and production of ozone (Shen et al., 2017a). Variations of these local meteorological variables depend on the dominant weather systems (Han et al., 2018b; Leung et al., 2018).

To have a comprehensive and quantitative understanding of how weather influences ozone pollution

in China is the primary motivation of this study, in which, we aim to quantify the impacts of meteorology, specifically the dominant synoptic weather systems and the key meteorological variables, on daily variations of surface ozone in eastern China, including the three representative megacity clusters, BTH, YRD, and PRD. Surface ozone in China was not regularly and systematically monitored until 2012 when real-time hourly ozone data were available online from China Ministry of Ecology and

Environment (MEE) (http://www.mee.gov.cn/) (T. Wang et al., 2017). Owing to the limitation of in situ measurement, there is a lack of a long-term assessment on the synoptic influence on ozone pollution.

The ground ozone observations from MEE covering 2013-2018 period are used. First, we characterize the seasonal variations of surface ozone in eastern China and the interannual changes

during 2013-2018 in summer (June-August), the season of interest in this study. Second, we search for a



linkage between the daily variation of surface ozone and the local and synoptic meteorological factors

statistically and develop a multiple linear regression (MLR) model based on the linkage. Third, we

examine the sensitivity of daily surface ozone to the variation in synoptic weather systems. Considering

the complexity of the synoptic meteorology in eastern China (Ding et al., 2017; Han et al., 2018b), we

100 employ an objective clustering technique, the self-organizing map (SOM), to identify the predominant

synoptic weather patterns (SWPs). In the following sections, we introduce the data and methods in

section 2. The seasonal and interannual variations of surface ozone in eastern China are characterized in

section 3. Section 4 illustrates the linkage between ozone variability and meteorology on both local and

synoptic scales, while section 5 describes sensitivity of surface ozone to various typical SWPs over the

105 entire eastern China. Finally, we discuss our results and draw conclusions in section 6.

## 2 Data and methods

### 2.1 Surface ozone observations and meteorological data

Hourly surface ozone measurements from the MEE observation network averaged over stations in each

110 city were used in the study. The measurements were downloaded from http://beijingair.sinaapp.com/,

which were previously archived at http://pm25.in, a mirror of data from the official MEE publishing

platform (http://106.37.208.233:20035/). The network covers 63 cities in eastern China (110-130°E, 20-

42°N) in 2013, increasing to 118 in 2014 and 185 during 2015-2018. Locations of the 185 cities are

shown in Figure 1, including 13, 26, and 9 cities, respectively, in BTH, YRD, and PRD.


 The National Centers for Environmental Prediction (NCEP) Final (FNL) Operational global analysis

data during the same period were acquired from https://rda.ucar.edu/datasets/ds083.2/. The data are

available on $1° \times 1°$ grids every 6 hours for the surface and at 26 layers from 1000 to 10 hPa. We made

daily averaged pollution and meteorological data in summer from 2013 to 2018.






Using inverse distance weighting (Tai et al., 2010), we interpolated the daily city measurements onto the FNL grid to produce continuous gridded data. Ozone at each FNL grid was calculated with a weighted average of the concentration in the cities within a search distance ($d_{max}$) from that grid, following the equation:

$$z_j = \frac{\sum_{i=1}^{n_j} (1/d_{i,j})^k z_i}{\sum_{i=1}^{n_j} (1/d_{i,j})^k} \qquad (1)$$


where $z_j$ is the calculated ozone at grid $j$, $z_i$ is the observed ozone in city $i$, $d_{i,j}$ is the distance between city $i$ and the center of grid $j$, $n_j$ is the number of the cities within $d_{max}$ from grid $j$ ($d_{i,j} \leqslant d_{max}$), $k$ is a parameter measuring the influence of distance on the target grid. We used 2 for $k$, and 1-degree distance in latitude-longitude grid for $d_{max}$ in the interpolation. The generated gridded ozone data cover most of

the mainland in eastern China (Figure 1c). The measurements interpolated to the grids were used in this study, unless stated otherwise.

## 2.2 Development of a prediction model of surface ozone

MLR is an effective and widely-used way to describe the relationship between meteorology and air

quality and thus to help prediction of air quality (Shen et al., 2015; Otero et al., 2016; K. Li et al., 2019). MLR establishes a linear function between a scalar response and the explanatory variables. In this study, we applied a stepwise MLR model to quantitatively correlate daily surface ozone in eastern China and meteorology in summer. Considering the combined effect of meteorology at various scales, we used both local meteorological variables and synoptic circulation factors as predictors following Shen et al.

(2017b), who showed that, comparing with regression models only considering local meteorology, adding the synoptic factors in a MLR can significantly improve the model performance. The MLR model takes the following form:

$$\hat{Y} = b + \sum_{i=1}^{K_1} \alpha_i X_i + \sum_{j=1}^{K_2} \beta_j S_j \qquad (2)$$





where $\hat{Y}$ is the predicted value of surface ozone, $b$ is the intercept term, $X_i$ is the local meteorological

variables with a total number of $K_1$, $S_j$ is the synoptic meteorological factors with a total number of $K_2$,

and $\alpha_i$ and $\beta_j$ are the regression coefficients. We used 10 local meteorological variables ($K_1$=10), i.e.,

relative humidity at 2 m (RH2m), cloud fraction (CF), temperature at 2 m (T2m), planetary boundary

layer height (PBLH), zonal wind at 850 hPa (U850), meridional wind at 850 hPa (V850), vertical wind

at 850 hPa (W850), wind speed at 850 hPa (WS850), geopotential height at 850 hPa (HGT850), and sea

level pressure (SLP), all of which were identified significantly ($p<0.05$) correlated to the daily

variations of surface ozone in part of eastern China, as shown in Figure 2. Cloud fraction retrievals at 1º

×1º grids were from the spaceborne Atmospheric Infrared Sounder (AIRS) instrument (AIRS3STD

daily product, https://disc.gsfc.nasa.gov/). The other 9 local meteorology were from FNL data (section

2.1). We computed the anomalies of meteorological variables and ozone on a given day by taking the

difference between the value of a given meteorological variable (or ozone) on that day and the mean

value of the variable (or ozone) in that month. Thus, all the data were detrended and the influences of

meteorology on the ozone variability on longer time scales (trends, and annual and seasonal variations)

were generally removed. Any anomaly of a variable (or ozone) divided by its corresponding monthly

mean is referred as relative anomaly of that variable (or ozone).


We also used two synoptic factors generated from the singular value decomposition (SVD) of the

spatial correlations between surface ozone and local meteorological variables in eastern China (Shen et

al., 2017b). The SVD approach effectively extracted representative signals from the spatial distribution

of the correlation coefficients. The extracted information was then used to characterize the spatial

patterns of the meteorological variables at a synoptic scale by inversing SVD. For each of the FNL grids

in eastern China, we constructed the synoptic circulation factors as follows. First, we calculated the

correlation coefficients between daily surface ozone at that grid and each of the 10 meteorological





variables at all the grids in eastern China in summer during 2013-2018. For example, the correlations

for the grid of Nanjing are shown in Figure S1, which indicates that surface ozone in Nanjing is

correlated to the meteorology in the surrounding regions. We made a matrix $A$ that consists of the

correlation coefficients for that grid with elements of 21 (numbers of grids in longitude) $\times$23 (numbers

of grids in latitude) $\times$10 (numbers of the local meteorological variables). Second, to fit the

decomposition, we aligned the dimension of longitude-latitude into one column and reshaped matrix $A$

into a 483 (longitude$\times$latitude)$\times$10 two-dimensional matrix $F$. The SVD decomposed $F$ used the

equation:

$$F=ULV^{T} \qquad (3)$$

where $U$ is 483$\times$10 matrix, $L$ is a 10$\times$10 diagonal matrix with non-negative numbers on the diagonal, $V$

is also a 10$\times$10 matrix. The columns of the three transformations together characterize SVD modes,

with 10 modes in total. Each column of $U$ represents the spatial weights of the SVD mode and each

column of $V$ represents the variable weights in the SVD mode. The spatial and variable weights of the

first two SVD modes for the Nanjing grid are shown in Figure S2. The pattern of the spatial weight of

the first SVD mode for the Nanjing grid (Figure S2a) is similar to the pattern of the correlations

between surface ozone and relative humidity (Figure S1a) and cloud fraction (Figure S1b). The first

SVD mode is more correlated to relative humidity and cloud fraction than other variables (Figure S2b).

Therefore, the first SVD mode for Nanjing grid is related to chemical processes of ozone. In contrast,

the second SVD mode for Nanjing grid is more related to transport than chemical processes (Figure

S2d). Third, we assigned the anomalies of the daily mean values of the 10 local meteorological

variables in eastern China to a 552 (days in summer of 2013-2018) $\times$21 (longitude)$\times$23 (latitude)$\times$10

(meteorology) four-dimensional matrix $M$. At each grid, we normalized the time series of each variable

to zero mean and unit standard deviation. Then, the magnitude of each SVD mode for every day $t$ was

calculated by inversing SVD:





$$S_{k,t} = U_k^T M_t V_k \qquad (4)$$

where $U_k$ and $V_k$ respectively are the $k^{th}$ columns of $U$ and $V$, $S_{k,t}$ is a scalar depicting the magnitude of

the $k^{th}$ SVD mode. $S_{k,t}$ refers to a newly produced meteorological field and reflects the synoptic factors

related to ozone variability. We implemented the procedure at every FNL grid in eastern China. The first

two SVD modes can generally explain 55-85% of the total variance. They can respectively reflect the

dynamical or thermal characteristics in the synoptic meteorology (Shen et al., 2017b). Therefore, we

applied the primary two SVD modes in the MLR model ($K_2$=2).

We measured the relative importance of each of the meteorological variables to ozone by its relative

contributions to the total explained variance of the MLR model. The weight of each predictor ($w_i$) was

calculated from the normalized MLR coefficient ($z_k$):

$$w_i = \frac{z_k^2}{\sum_{k=1}^{12} z_k^2} \qquad (5)$$

where $z_k$ is:


$$z_k = \frac{s_k}{s_y} c_k \qquad (6)$$

and $12$ is the number of all the predictors including 10 local and 2 synoptic factors (section 2.2). $c_k$ is

the regression coefficient, referring to $\alpha_i$ or $\beta_j$ in equation (2). $s_k$ is the standard deviation of a predictor,

$X_i$ or $S_j$ in equation (2). $s_y$ is the standard deviation of the observed daily surface ozone.

**2.3 Classification of the synoptic weather patterns**

Weather classification is a well-established tool to characterize atmospheric processes at multiple scales

and further to study air pollution-weather relationship (Han et al., 2018b). The methods for weather

classification can be generally categorized into three groups: subjective, mixed, and objective,

depending on the automatic degree during the classification process (Huth et al., 2008). The methods





can also be categorized in more detail according to the basic features of each classification algorithm
(Philipp et al., 2014). Depending on the study domain and research objectives, different meteorological
variables including geopotential height, mean sea level pressure, and zonal and meridional winds are
used for the classification.

SOM, an artificial neural network method with unsupervised learning (Kohonen, 1990; Michaelides
et al., 2007), is widely used in cluster analysis in atmospheric sciences (Jiang et al., 2017; Liao et al.,
2018; Stauffer et al., 2018) because of its superiorities over other algorithms (Liu et al., 2006; Jensen et
al., 2012). SOM performs a nonlinear projection from the input data space to a two-dimensional array of
nodes objectively. Each node is representative of the input data. SOM allows missing values in the input
data and can effectively visualize the relationships between different output nodes (Hewitson and Crane,
2002).

The FNL geopotential height fields (section 2.1) at 850 hPa can well capture the synoptic circulation
variations over eastern China (Han et al., 2018b). In this study, we used geopotential height at 850 hPa
in 2013-2018 as the input for SOM. Each of the SOM output nodes corresponds to a cluster of SWPs.
Finally, we identified six predominant SWPs over eastern China in summer. All days in summer of
2013-2018 were included in the clustering results.

**3 Seasonal and interannual variations of surface ozone in eastern China**

Figure 3 and Figure 4, respectively, show the seasonal and interannual variations of regional mean
surface ozone concentrations in eastern China and the three subregions (BTH, YRD, and PRD) during
2013-2018. Among $n$ cities with air quality monitoring in a given region, if ozone levels exceed the
national air quality standard in $m$ cities, we defined the ratio of $m$ to $n$ as the regional exceedance





probability of ozone (Figure 3c). Higher regional exceedance probability implies ozone pollution over

wider surface areas in that region. Primary pollutant (Figure 3d) is defined in the Air Quality Index

(AQI) system, in which, AQI for an individual air pollutant is calculated based on the concentrations of

the pollutant. When the individual AQI of a pollutant on a day is both above 50 and the largest among

all the pollutants on that day this pollutant is defined as the primary pollutant on that day.

On the regional average, the seasonality of daily mean ozone is similar to that of daily maximum 8-

hour average (MDA8) ozone in eastern China, as well as the three subregions (BTH, YRD, and PRD)

(Figures 3a and 3b). In BTH, both daily mean and MDA8 have a unimodal seasonal pattern and peak in

June, being 99.5 and 158.4 μg m$^{-3}$, respectively. The extremely high ozone in June leads to a

simultaneous seasonal maximum in both probability of the regional exceedance (46.9% of the cities

with ozone measurements in BTH) and primary pollutant (68.7% of the days in June) (Figures 3c and

3d). The seasonal peak of surface ozone in BTH mainly results from enhanced photochemistry due to

stronger solar radiation and lower humidity (Hou et al., 2014). Surface ozone over YRD reaches a

seasonal maximum in May (82.6 and 127.7 μg m$^{-3}$, respectively, for daily mean and MDA8 ozone),

earlier than that over BTH. While the seasonal peak over PRD occurs the latest in October (71.5 and

118.1 μg m$^{-3}$, respectively, for daily mean and MDA8 ozone). Although the temperature is higher in

summer than in the other seasons, the EASM brings more cloudy weather, stronger convection, and

clearer air from the oceans, weakening the production and accumulation of surface ozone over YRD and

PRD (Hou et al., 2015; S. Li et al., 2018). The pre-monsoon and post-monsoon peak of surface ozone

were also found in YRD and PRD, respectively (He et al., 2008; T. Wang et al., 2009).


     On the regional and seasonal average, daily mean and MDA8 ozone over eastern China in summer

are 73.1 and 114.7 μg m$^{-3}$, respectively. Among the three city clusters, summertime surface ozone is




highest in BTH (88.3 and 143.7 μg m$^{-3}$, respectively, for daily mean and MDA8 ozone), second highest

in YRD (72.9 and 114.7 μg m$^{-3}$), and lowest in PRD (51.0 and 91.9 μg m$^{-3}$) (Figures 3a and 3b). These

regional differences among the three city clusters appear similar to these in the Ozone Monitoring

Instrument (OMI) tropospheric column ozone (Figure 1). The regional exceedance probability of ozone

over eastern China reaches 17.7% in summer, accompanied with a high percentage (45.6%) of ozone

being the primary pollutant. Among the three subregions, the regional exceedance probability of ozone

(35.1%) and the probability of ozone being the primary pollutant (55.8%) are largest in BTH.


A rapid increasing trend in summertime surface ozone over China after 2012 was observed in recent

studies (Lu et al., 2018; Silver et al., 2018; Shen et al., 2019; K. Li et al., 2019). We examine the

regional mean trend over eastern China in daily, daytime (7:00-18:00), and nighttime (19:00-6:00)

means (Figure 4). Significant ($p<0.05$) summer increasing trends of approximately 3-6 μg m$^{-3}$ or 4-8%

per year are found over eastern China, BTH, and YRD during 2013-2018, while the increasing trend

over PRD during the period is insignificant ($p>0.05$). Silver et al. (2018) found the annual mean MDA8

ozone has increased significantly ($p<0.05$) at around 50% of the over 1000 stations across China from

2015 to 2017, with a median rate of 4.6 μg m$^{-3}$ year$^{-1}$. The increasing trend over eastern China was also

captured by the OMI satellite records of tropospheric ozone, reported by Shen et al. (2019). The

absolute increasing trend (in a unit of μg m$^{-3}$) in daytime is higher than that in nighttime, whereas the

relative increasing trend (in a unit of %) in daytime is lower than that in nighttime (Figures 4e-4h vs.

Figures 4i-4l). The increasing ozone trend over China may result from both meteorology and

anthropogenic emissions. K. Li et al. (2019) suggested the ~40% decrease of fine particulate matter

(PM$_{2.5}$) is the primary reason for the increasing trend of surface ozone in summer during 2013-2017, as

the aerosol sink of hydroperoxy radicals was weakened and thus ozone production was enhanced.

Figure 4 demonstrates a strong increase in summertime surface ozone over BTH from 2016 to 2017,





which is probably related to the hot extremes in 2017 (Herring et al., 2019). The sudden decline in summertime surface ozone over PRD from 2016 to 2017 (Figure 4d) is likely associated with the extremely heavy precipitation in 2017 (Herring et al., 2019).


## 4 Meteorological drivers for summertime surface ozone in eastern China

Meteorological factors can individually or integrally modulate surface ozone concentration through their impacts on relevant chemical, dynamical, and thermal processes in the atmosphere. Figure 2 shows a simple way to examine the overall effect of each of the meteorological variables statistically by

correlating surface ozone with a selected set- of local meteorological variables during 2013-2018 summer. Among all the meteorological variables, relative humidity shows the highest correlation with surface ozone in eastern China on regional mean. The correlation map of cloud fraction is similar to that of relative humidity (Figures 2a and 2b). The correlation of temperature with ozone is higher in the north than in the south over eastern China (Figure 2c), which may due to lower humidity in the north.

Meridional wind at 850 hPa is positively correlated to surface ozone in the north but negatively in the south (Figure 2f). Higher meridional wind brings clean and humid marine air to the south, while it transports ozone and its precursors from the south to the north. All the meteorological variables are not independent with each other. For example, relative humidity is strongly correlated with cloud fraction. A higher relative humidity is usually associated with more fractions of clouds, which can slow the

photochemical production of surface ozone. The impacts of relative humidity on surface ozone are mainly through the chemical processes. In addition, higher relative humidity may somewhat be linked with larger atmospheric instability, favoring the dispersion of surface ozone (Camalier et al., 2007). Overall, the meteorological variables that are related to photochemistry processes (relative humidity, cloud fraction, and temperature) have more significant correlation than transport-related variables

(zonal, meridional, and vertical winds and wind speed) (Figure 2), implying greater effects of chemical



process than physical transport. S. Li et al. (2018) also suggested the chemical process is the uppermost factor controlling surface ozone levels over eastern China in summer.

Combining the effects of different meteorological variables, we applied the MLR model using
predictors of both local and synoptic factors (section 2.2) to simulate summertime daily surface ozone in eastern China. The MLR model was developed using the observation data in 2013-2017 and evaluated with observation data in 2018. The MLR model performs strongly as it can explain 30-65% variations in the observed surface ozone concentrations in 2013-2017, yielding a regional mean coefficient of determination ($R^2$) of 46% (Figure 5c). Geographically, the model performs better in the
south ($R^2$=0.52 in YRD and $R^2$=0.54 in PRD) than in the north ($R^2$=0.44 in BTH) (Figure 5c). In the validation period, the model also shows strong performance ($R^2$=0.38 in eastern China) (Figure 5d). Moreover, we simulated surface ozone considering only the local meteorological variables (Figure 5a) and only the synoptic factors (Figure 5b) in the MLR model. Compared with the two simulations (Figures 5a and 5b), the model performance is overall improved in areas in eastern China when both
local and synoptic factors are considered (Figure 5c). Shen et al. (2017b) found that, compared with the MLR model describing monthly $PM_{2.5}$ in the United States driven by only the local meteorology, the inclusion of the synoptic factors in the MLR model increases $R^2$ from 34% to 43%. In addition, we conducted the stepwise MLR model using local and synoptic meteorology without detrending the input data. The results show that meteorology can explain 39% of the increasing trend in the regional mean of
summertime surface ozone over eastern China from 2013 to 2018, and the explained variance is 23%, 53%, and 57% for BTH, YRD, and PRD, respectively (Figure S3).

We applied the MLR model to identify the dominant meteorological drivers for ozone variability (section 2.2). Synoptic factors diagnosed by SVD are the most pronounced drivers in ~45% areas of





eastern China and contribute to 30-60% of the meteorological effects on surface ozone over these locations (Figure 6b). The regional mean contributions of the synoptic factors are 37% over eastern China and 41% over BTH, YRD, and PRD (Figure 6b). Among the local meteorology, relative humidity is dominant over ~48% areas of eastern China, mainly in the central and the southern regions including YRD and PRD (Figure 6c). Relative humidity is estimated to account for ~30% of the meteorological impacts on daily surface ozone variation in YRD and PRD on the regional scale (Figures 7c and 7d), although at a city scale in PRD, Zhao et al. (2016) suggested that sea level pressure is the most significant variable for MDA8 ozone in Hong Kong. Air temperature is the most important local meteorological variable in ~15% areas of eastern China, specifically in the north including BTH (Figure 6c). Temperature is estimated to account for 20% of the meteorological impact in BTH (Figure 7b). The importance of temperature to surface ozone over BTH was also suggested by Chen et al. (2019). Previous studies found that temperature and relative humidity showed pronounced impact on ozone in the north and south of the eastern US, respectively (Camalier et al., 2007; Porter et al., 2015). In Europe, Otero et al. (2016) suggested that temperature is the most important local meteorological driver over a major part of Europe. On regional average, the second most important meteorological variable for the daily surface ozone variation in eastern China, BTH, YRD, and PRD is temperature, relative humidity, geopotential height at 850 hPa, and meridional wind at 850, respectively (Figure 7).

**5 Synoptic impacts on summertime surface ozone in eastern China**

In the last section, we have shown that both local and synoptic meteorological factors are important for surface ozone in eastern China. The synoptic factors used there were extracted via an inversing SVD process and do not stand for specific weather systems. In this section, we will further show how the specific synoptic weather systems influence surface ozone in eastern China by looking into the typical SWPs. Atmospheric circulations over eastern China in summer are largely regulated by the evolution of





the components of EASM, for instance, the western Pacific subtropical high (WPSH), the subtropical

westerly jet, the Meiyu front, and the Southwest Vortex (Ding and Chan, 2005). Among these systems,

the WPSH can largely modulate the seasonal migration of the rain belt over eastern China. Typhoon is

also an influential weather system, especially on the southeast coastal regions. The main features of the

synoptic circulations over eastern China during 2013-2018 can be represented by six predominant

SWPs (Figures 8-13), which were identified by an objective approach, SOM (section 2.3). The

occurrence frequency of these SWPs is shown in Figures 8-13. We name the six SWPs by their

dominant weather systems or prevailing wind, including Pattern 1 featured southwesterly wind (P1 or

PSW), Pattern 2 featured Southerly wind (P2 or PS), Pattern 3 featured Northeast Cold Vortex (P3 or

PNECV), Pattern 4 featured a weak cyclone (P4 or PWC), Pattern 5 featured strong WPSH (P5 or

PSWPSH), and Pattern 6 featured typhoon systems (P6 or PTC) (Table 1).


To compare the differences of meteorological conditions among the six SWPs, we calculated the

daily EASM index (EASMI) and WPSH index (WPSHI) representing the strength of EASM and WPSH

respectively. The two indexes were normalized to zero mean and unit standard deviation. The averaged

anomalies of the normalized indexes under each SWP are shown in Figures 8-13 and Table 1. The

EASMI is a shear vorticity index defined as the difference of the regional mean zonal wind at 850 hPa

between 5-15°N, 90-130°E and 22.5-32.5°N, 110-140°E in B. Wang and Fan (1999) recommended by B.

Wang et al. (2008). The WPSHI is defined by the accumulative enhancement of geopotential height

above the WPSH characteristic isoline (5880 gpm at 500 hPa) averaged over the area north to 10°N. The

WPSHI is adopted by the National Climate Center in China (https://cmdp.ncc-cma.net) in the

monitoring and diagnosis of the atmospheric circulation. Using the WPSHI, Zhao and Wang (2017)

found a significant correlation between the WPSH and the first empirical orthogonal function (EOF)

pattern of surface ozone in China. Moreover, we used the averaged anomalies of the meteorological


variables in a SWP to describe that SWP. We used the averaged ozone anomaly (in μg m$^{-3}$) (Figures 8-

13) and the averaged relative ozone anomaly (the ozone anomaly divided by the monthly ozone mean,

in %) (Figure 14) under a SWP to assess the influence of that SWP on ozone (Han et al., 2018b).

Furthermore, a common index for air stagnation (Horton et al., 2012) is used to assess the impact of air

stagnation on surface ozone. For each FNL grid, when daily average wind speed at 10 m, daily average

wind speed at 500 hPa, and the daily total precipitation are respectively less than 3.2 m s$^{-1}$, 13 m s$^{-1}$, and

1 mm, the day is considered as a stagnant day for that grid. The National Oceanic and Atmospheric

Administration (NOAA) Climate Prediction Center (CPC) precipitation data

(https://www.esrl.noaa.gov/psd/data/gridded/data.cpc.globalprecip.html) were used in the calculation of

the air stagnation index.

The characteristics of the six SWPs and their impacts on surface ozone are briefly summarized in

Table 1. PSW (P1) is the most common circulation pattern occurring in 25% days of summer during

2013-2019 (Figure 8b). Characterized with weak EASM conditions, PSW is dominated by an

anomalous anticyclone located in the southeast of eastern China (Figure 8e). In PSW, the enhanced

meridional wind brings clear marine air to the south of eastern China (Figure 8j), where the meridional

wind is significantly correlated to surface ozone (Figure 2f). The enhanced zonal wind from the

anomalous anticyclonic circulation (Figure 8e) increases the ozone export from the south of eastern

China (Yang et al., 2014). The negative anomalies of temperature (Figure 8g), and positive anomalies of

relative humidity (Figure 8f) and cloud fraction (Figure 8h) in the south are unfavorable for

photochemical. In consequence, PSW reduces ozone levels in the south (Figure 8c) by enhancing the

dispersion and suppressing the production of ozone. Negative anomalies of -1.5 (-2.4%) and -6.6 μg m$^{-3}$

(-13%) in regional mean ozone are respectively observed over YRD and PRD (Figures 8c and 14a). In

contrast, the lower cloud fraction (Figure 8h) and higher temperature (Figure 8g) in the north stimulate



ozone production. Surface ozone over BTH increases by 3.4 μg m$^{-3}$ (3.6%) from the regional mean in

PSW (Figures 8c and 14a).

PS (P2) is the second frequent SWP (Figure 9b), characterized with strong EASM and weak WPSH

(Figure 9a). In PS, FNL data illustrate frequent stagnation events (Figure 9l), low humidity (Figure 9f),

and low cloud fraction (Figure 8h) over most of eastern China. In contrast to PSW, the zonal wind has

negative anomalies (Figure 9e) in PS, reducing ozone export from the south of eastern China. Overall,

an increase of 1.1 μg m$^{-3}$ (1.7%) in the regional mean ozone concentrations in eastern China is observed

in PS (Figures 9c and 14b).

PNECV (P3) is a typical pattern for Meiyu, an important climate phenomenon over the middle and

lower reaches of the Yangtze River during early June to mid-July (Figure 10a), characterized by

persistent rainfall (Ding and Chan, 2005). Under a combined effect of the Northeast Cold Vortex and the

WPSH, Meiyu front forms and maintains over YRD (He et al., 2007). Meiyu in PNECV increases

relative humidity (Figure 10f) and decreases air stagnation (Figure 10l) over YRD. Consequently,

PNECV reduces surface ozone concentrations by 1.3 μg m$^{-3}$ (1.7%) over YRD (Figures 10c and 14c).

Meantime, more sunny days with high temperature (Figure 10g) and low moisture (Figure 10f) occur in

the north to YRD, affected by the northwesterly and downward airflows from the Northeast Cold Vortex

(Figure 10a). As a result, positive ozone anomalies are observed in the regions north of YRD (Figure

10c).

PWC (P4) features the weakest WPSH, when a weak extratropical cyclone locates over the east of the

mainland China (Figure 11a). The extratropical cyclone is probably formed by the eastward movement

of the Southwest Vortex or the transition from typhoon. Pushed by the cyclone, the WPSH retreats (Y.





Li et al., 2018). The weak pressure gradient over the mainland of eastern China (Figure 11a) in PWC results in more stable weather conditions. The anomalies of the meteorological variables in PWC show opposite spatial patterns to those in PSW (Figure 8 vs Figure 11). With the favorable meteorological conditions except temperature, PWC enhances ozone over the south, with increased regional mean values of 5.2 $\mu$g m$^{-3}$ (7.5%) over YRD and 6.7 $\mu$g m$^{-3}$ (11.8%) over PRD (Figures 11c and 14d).

PSWPSH (P5) occurs in late summer (Figure 12a), when Meiyu breaks in the Yangtze River and the rain belt jumps to North China (Ding and Chan, 2005). In PSWPSH, the WPSH is the strongest and extends westward the mostly (Figure 12a). Thus, relative humidity is lower than the seasonal mean over YRD and higher than the seasonal mean over BTH (Figure 12f). Meantime, stable weather conditions occur more frequently over YRD (Figure 12l). Therefore, ozone accumulates over YRD in PSWPSH with a regional mean enhancement of 1.8 $\mu$g m$^{-3}$ (2.5%) (Figures 12c and 14e). Surface ozone decreases by 0.8 (1.4%) and 5.0 $\mu$g m$^{-3}$ (8.9%) respectively over BTH and PRD under this SWP (Figures 12c and 14e).

PTC (P6) is a typical typhoon weather pattern that is over the southeast coast of the mainland China (Figure 13a). Forced by typhoon, the WPSH in PTC migrates further north than under the other SWPs. The typhoon system brings clear and moist marine air to coastal cities in eastern China, reducing surface ozone by 6.8 $\mu$g m$^{-3}$ (9.2%) over YRD (Figures 13c and 14f). Shu et al. (2017) identified that SWPs like PTC can lead to clean $PM_{2.5}$ episodes in YRD. However, the cyclonic circulation enhances ozone transport from the central part of eastern China to the downwind regions in the south including PRD. The collective effect of higher temperature, lower humidity, and heavier downdrafts, PTC increases surface ozone in PRD by 7.9 $\mu$g m$^{-3}$ (15.5%) (Figures 13c and 14f). Lam et al. (2018) found ozone increases by 16.8 $\mu$g m$^{-3}$ at urban stations in Hong Kong of PRD, when the synoptic circulation





controlling PRD is featured typhoon in the vicinity of Taiwan, similar to PTC. They also suggested that

this SWP is associated with the interannual variations of ozone pollution in Hong Kong.

We further compared the SWPs analysis with that from the MLR model discussed in section 4. We

evaluate the performance of the MLR model under the six SWPs based on the predicted (Figures 8d, 9d,

10d, 11d, 12d, and 13d) and observed (Figures 8c, 9c, 10c, 11c, 12c, and 13c) ozone anomalies. The

comparison shows that the ozone anomalies predicted by the MLR have spatial variations and

magnitudes similar to those in the observations under each of the SWPs. The MLR model can well

capture the ozone anomalies under the six predominant SWPs (Figures 8-13). For example, the negative

ozone anomaly over PRD under PSW (P1) featured weak EASM (Figures 8c vs. 8d), the negative ozone

anomaly over YRD under PNECV (P3) featured Meiyu (Figures 10c vs. 10d) and the positive ozone

anomaly over PRD caused by PTC (P6) featured typhoon (Figures 13c vs. 13d). Since the MLR model

only considers the meteorological influence on surface ozone, the consistency between the regression

and the clustering results suggests that the mean observed ozone anomalies under a SWP can adequately

reflect the sensitivity of ozone to meteorology. The noise of day-to-day variations of chemistry and

emissions in the surface ozone data can be largely removed by long-term average of ozone anomalies

under a SWP from the big data set of surface ozone (Han et al., 2018b).

## 6 Discussion and conclusions

Meteorology can influence surface ozone variability on different time scales, from long-term trends to

sub-daily scale. Based on surface ozone observations in eastern China during 2013-2018 from MEE, we

characterized the seasonal and interannual variations of surface ozone in eastern China. The

measurements show that surface ozone pollution in the study region is severest in summer and the

severity goes in a rapid increasing trend during the study period. We then focused on the meteorological



influence on the daily variability of summertime surface ozone in eastern China. We took daily

anomalies of meteorological and ozone values to remove the variabilities on longer time scales in these

datasets. We estimated the local and synoptic meteorological impacts on daily variability of surface

ozone using a MLR model and a SOM clustering technique. Synoptic weather factors identified by the

SVD analysis were combined with local meteorological variables to drive the MLR model. The

regression model suggests that on regional average, meteorology can explain 46% variations in the

summertime daily surface ozone in eastern China, with an explained variance of up to 65% over some

locations (Figure 5c). The model also shows that meteorology contributes to 39% of the increasing trend

in the regional mean of summertime surface ozone over eastern China from 2013 to 2018.

Exploiting the MLR model, we also identified the key meteorological variables that are mostly

responsible for variations of summertime surface ozone in eastern China during 2013-2018. On regional

average, the synoptic circulation factors constructed by the SVD analysis were estimated to contribute

to 37% of the meteorological impact over eastern China and 41% over BTH, YRD, and PRD (Figure

6b). Among the local meteorological variables, relative humidity is the foremost over most locations in

the center and south of eastern China including YRD and PRD, while temperature is more important in

the north including BTH (Figure 6c).

We assessed the impacts of the dominant synoptic weather systems on surface ozone using cluster

analysis. Employing the SOM, the summer synoptic circulations over eastern China during 2013-2018

were objectively classified into six predominant SWPs (Figures 8-13). The six SWPs control the

variations of the key meteorological variables and thus impact the transport and production of ozone. As

the predominant meteorological controlling variables of surface ozone vary greatly in space (Figures 2

and 6), strong differences are found in surface ozone concentrations under every SWP between northern





and southern parts of eastern China or between eastern and western parts of eastern China (Figures 8-14). Daily surface ozone over some regions in eastern China can maximally increase or decrease by 8 μg m$^{-3}$ or 16% impacted by the dominant SWP.

Among the six SWPs, the SWP (PS) featured southerly wind, strong EASM and weak WPSH (Figure 9a), and the SWP (PWC) featured a weak extratropical cyclone and the weakest WPSH (Figure 11a) tend to increase the regional mean surface ozone in eastern China (Figures 9c and 11c). For specific regions in eastern China, PS and PWC statistically enhance ozone in YRD and PRD. However, mean negative ozone anomalies are observed over some locations in PS and PWC, such as BTH in PWC.

In contrast, the other four SWPs (namely, PSW, PNECV, PSWPSH, and PTC) tend to reduce regional mean surface ozone in eastern China. PSW is a SWP featured southeasterly wind and weak EASM (Figure 8a) and it leads to overall ozone reduction in the south of eastern China including YRD and PRD (Figure 8c). When eastern China is influenced by the Northeast Cold Vortex, the WPSH, and the Meiyu front (Figure 10a), PNECV tends to reduce ozone over YRD (Figure 10c). Whereas, ozone likely increases over some subregions in eastern China in the four SWPs. PSWPSH featured the strongest and the most extensive WPSH (Figure 12a) can enhance ozone over YRD, although it may reduce regional mean ozone over eastern China (Figure 12c). When the atmospheric circulation is controlled by the typhoon systems with their centers around Taiwan, southeast to the mainland of China (Figure 13a), PTC reduces ozone in YRD, while enhances ozone in PRD (Figure 13c).

This study provides some new insights on the relationship between meteorology and air pollution, by untangling the complex response of surface ozone to different SWPs and meteorological variables. The most significant meteorological variables for surface ozone in eastern China were identified regionally,





which was rarely investigated by previous studies (Gong et al., 2018; Zhan et al., 2018; Chen et al., 2019). Extending from previous studies, we quantified ozone anomalies in eastern China resulting from the prominent synoptic weather systems such as the WPSH (Shu et al., 2016; Zhao and Wang, 2017),

the extratropical cyclones (Zhang et al., 2013; Liao et al., 2017), the Meiyu front, and typhoon (Jiang et al., 2015; Lam et al., 2018). These systems are important drivers for variations of air pollutants over eastern China (Ding et al., 2017). The relationship between weather and ozone is examined in one specific season, summer. Averaged ozone anomalies under a SWP in a relatively long term (2013-2018) was used to represent the ozone sensitivity to that SWP. This method is also applicable for a full year, as

it can remove the seasonal differences in the concentrations of pollution and the frequency of SWPs (Han et al., 2018b). No consideration of seasonal differences in ozone concentrations and meteorology can lead to biases (e.g. Zhang et al., 2013, 2016; Liao et al., 2017).

In this study, the developed MLR and cluster techniques can well describe the meteorological impacts

on the surface ozone variation in eastern China. Both regression and clustering analyses show strong performance, so they can be effective tools for air quality forecast. Many previous studies have reported the significance of local meteorology to the prediction of daily ozone in China (Zhao et al., 2016), however, few have included the meteorology at a synoptic scale. Here, we emphasized the synoptic role in the meteorological effects on surface ozone. The constructed synoptic factors by the SVD analysis

can be a useful predictor for short-term forecast of surface ozone. Regarding the time scale, this study focused on the day-to-day variations of surface ozone. Investigating the meteorological influences on a shorter time scale, such as diurnal variations, should be one of the directions for future work. The regression and clustering approaches can also be applied to project the potential effects of climate change on ozone variations in the future (Shen et al., 2017b). In the MLR regression analysis, we

focused on the meteorological effects without direct consideration of variations in emissions, assuming



emissions in a season are more or less constant. As ozone responses nonlinearly to variations in

meteorology, emissions, and chemistry (Wu et al., 2009), the developed MLR model cannot fully

describe the importance of meteorology to surface ozone variations. Therefore, future work is needed to

address the nonlinearity issue.


## Data availability

Surface ozone measurements were obtained from the public website of MEE

(http://beijingair.sinaapp.com/). The FNL meteorological data were acquired from NCEP

(https://rda.ucar.edu/datasets/ds083.2/). The OMI tropospheric column ozone monthly data were from

NASA Goddard Space Flight Center (https://acdext.gsfc.nasa.gov/Data_services/cloud_slice/).

## Author contributions

H. Han designed the study and performed the research. H. Han and L. Shu analyzed the data and

developed the model. H. Han and J. Liu wrote the manuscript with inputs from L. Shu, T. Wang, and H.

Yuan.

## Competing interests

The authors declare that they have no conflict of interest.

## Acknowledgements

We are grateful to MEE for the available air pollution data, to NCEP for the FNL meteorological data,

and to NASA Goddard Space Flight Center for the OMI tropospheric column ozone data. This research

is supported by the Chinese Ministry of Science and Technology under the National Key Basic

Research Development Program (2016YFA0600204, 2014CB441203) and by the Natural Science



Foundation of China (41375140, 91544230).

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



Table 1. Characteristics of the six predominant SWPs. Ozone anomalies observed in each SWP are shown in regional mean ± two times of the standard error of the mean. Anomalies of normalized EASMI and WPSHI in each SWP are shown to respectively represent the strength of EASM and WPSH. A higher EASMI (WPSHI) indicates a stronger EASM (WPSH).

| SWPs (Frequency) | Meteorological Conditions | | | Ozone Anomaly ($\mu g\ m^{-3}$) | | | |
|---|---|---|---|---|---|---|---|
| | Typical features | EASMI | WPSHI | EC | BTH | YRD | PRD |
| P1: PSW (25.0%) | Southwesterly wind | -0.7 | 1 | -1.3±0.2 (-2.9% ±0.2%) | 3.4±0.6 (3.6% ±0.6%) | -1.5±0.5 (-2.4% ±0.7%) | -6.6±0.5 (-13.0% ±0.9%) |
| P1: PS (20.8%) | Southerly wind | 0.6 | -0.4 | 1.1±0.2 (1.7% ±0.3%) | 1.4±0.7 (1.5% ±0.8%) | 1.0±0.6 (1.4% ±0.8%) | 1.1±0.7 (2.9% ±1.4%) |
| P3: PNECV (16.1%) | Northeast Cold Vortex and Meiyu | -0.9 | -0.9 | -1.1±0.2 (-1.2% ±0.3%) | -0.1±0.7 (0.6% ±0.8%) | -1.3±0.6 (-1.7% ±0.9%) | 1.6±0.8 (2.7% ±1.4%) |
| P4: PWC (15.4%) | Weak cyclone | 0.3 | -2.0 | 2.7±0.2 (4.6% ±0.3%) | -4.8±0.7 (-5.1% ±0.8%) | 5.2±0.7 (7.5% ±1.0%) | 6.7±0.9 (11.8% ±1.6%) |
| P5: PSWPSH (13.2%) | Strong WPSH | 0 | 2.2 | -0.4±0.2 (-0.9% ±0.3%) | -0.8±0.8 (-1.4% ±0.9%) | 1.8±0.6 (2.5% ±0.9%) | -5.0±0.9 (-8.9% ±1.6%) |
| P6: PTC (9.4%) | Typhoon | 1.6 | 0.1 | -0.7±0.3 (-0.1% ±0.4%) | -2.6±1.0 (-3.2% ±1.2%) | -6.8±0.9 (-9.2% ±1.3%) | 7.9±1.2 (15.5% ±2.2%) |




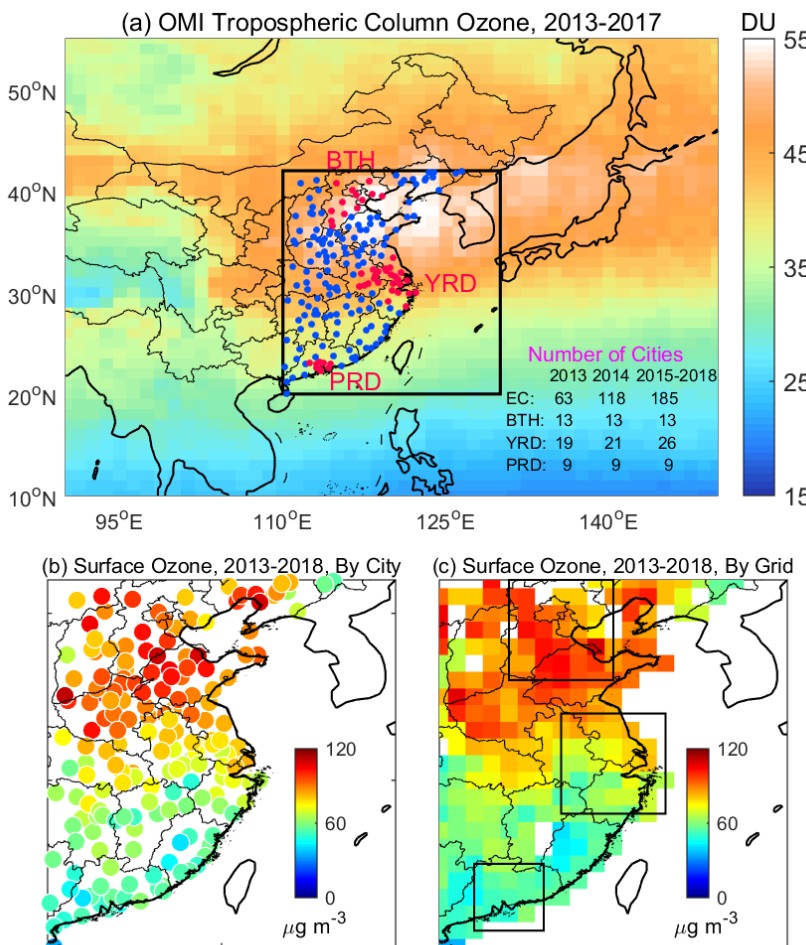

Figure 1. (a) Distribution of the cities (dots) with air quality monitoring in eastern China (EC, 110-130°E, 20-42°N, the boxed area) and the summer mean tropospheric column ozone (color shades, in Dobson units, DU) from the OMI satellite measurements during 2013-2017. Summertime mean surface ozone (in μg m$^{-3}$) over eastern China during 2013-2018 shown (b) by city and (c) by grid (see section 2.1). The red dots in (a) indicate the cities in BTH, YRD, and PRD. The three boxed areas in (c) indicate BTH (114-120°E, 36-42°N), YRD (117-123°E, 28-34°N), and PRD (112-116°E, 21-25°N). The unfilled grids in (c) are due to the lack of monitoring stations nearby. The OMI tropospheric column ozone monthly data at 1° latitude by 1.25° longitude were obtained from NASA Goddard Space Flight Center (https://acd-ext.gsfc.nasa.gov/Data_services/cloud_slice/).





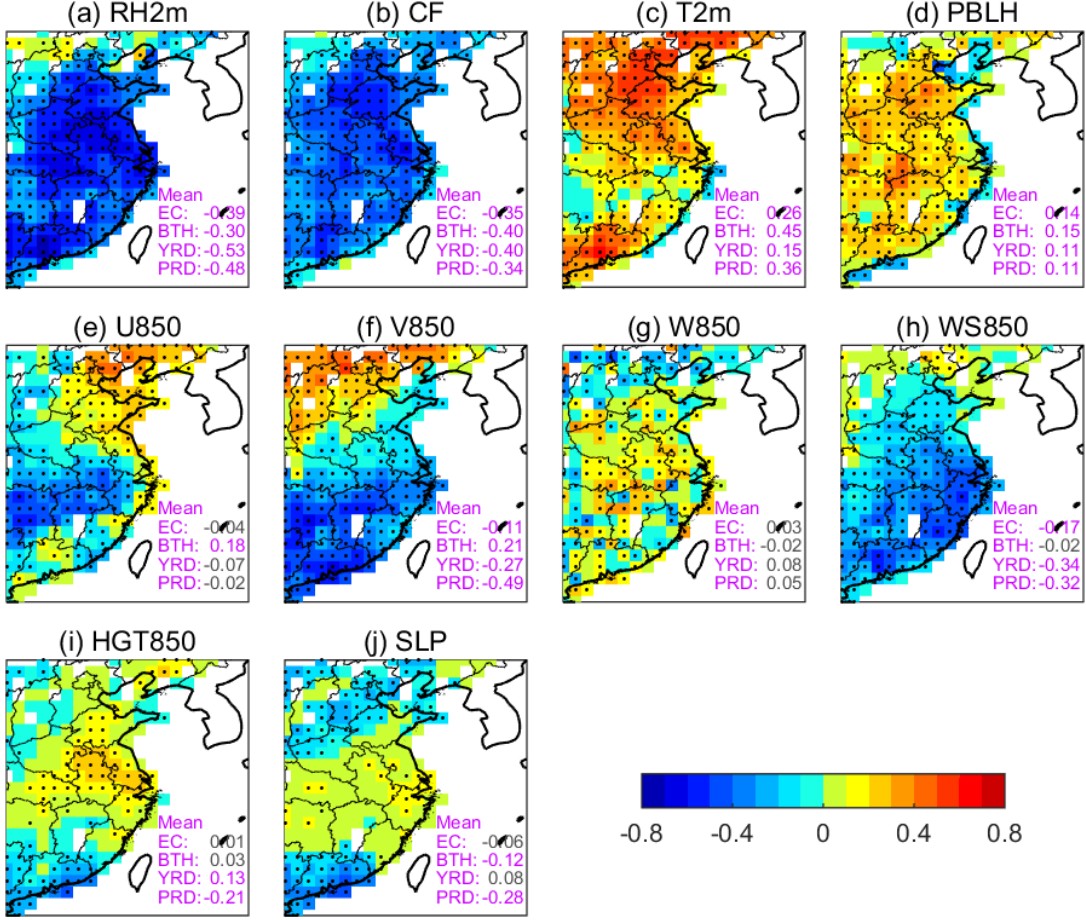

Figure 2. Correlation coefficients (*r*) between daily surface ozone and each of the meteorological variables in summer during 2013-2018. The black dot in a grid indicates that the correlation coefficient between daily surface ozone and the specific meteorological variable in the grid box is significant ($p<0.05$). The regional mean *r* is shown in the bottom right corner of each panel in purple if the *r* is significant ($p<0.05$) and in grey if the *r* is insignificant. The abbreviations are for relative humidity at 2 m (RH2m), cloud fraction (CF), temperature at 2 m (T2m), planetary boundary layer height (PBLH), zonal wind at 850 hPa (U850), meridional wind at 850 hPa (V850), vertical wind at 850 hPa (W850), wind speed at 850 hPa (WS850), geopotential height at 850 hPa (HGT850), and sea level pressure (SLP).



890

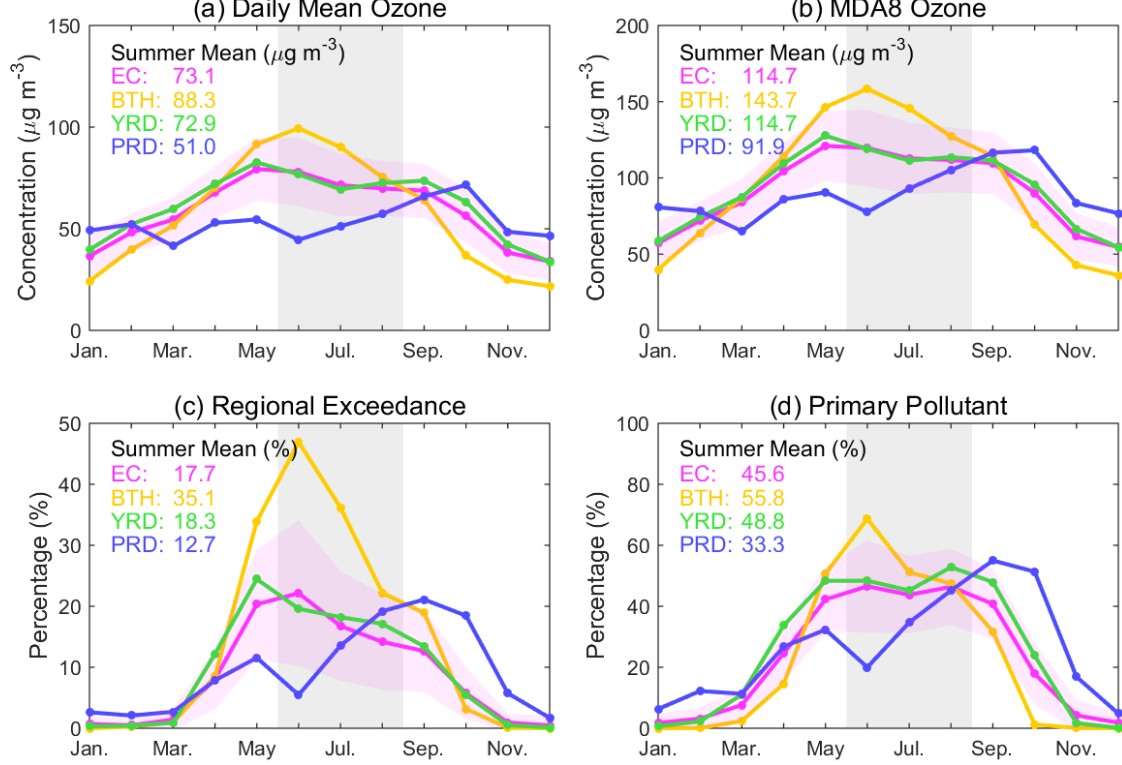

Figure 3. Seasonal variations of (a) daily mean surface ozone concentrations, (b) daily maximum 8-hour average surface ozone concentrations, (c) the regional exceedance probability of ozone, and (d) the probability of ozone being the primary pollutant. The values are regional means over eastern China (EC) and the three subregions (BTH, YRD, and PRD) in 2014-2017. The values were calculated from the observations of the corresponding cities. The pink shading area indicates the range of the ±50% the standard deviation of the corresponding variable in each panel for eastern China. The vertical shading over summer shows the season of interest in this study. Mean values for eastern China and the three subregions in summer are shown in the top left corner of each panel.



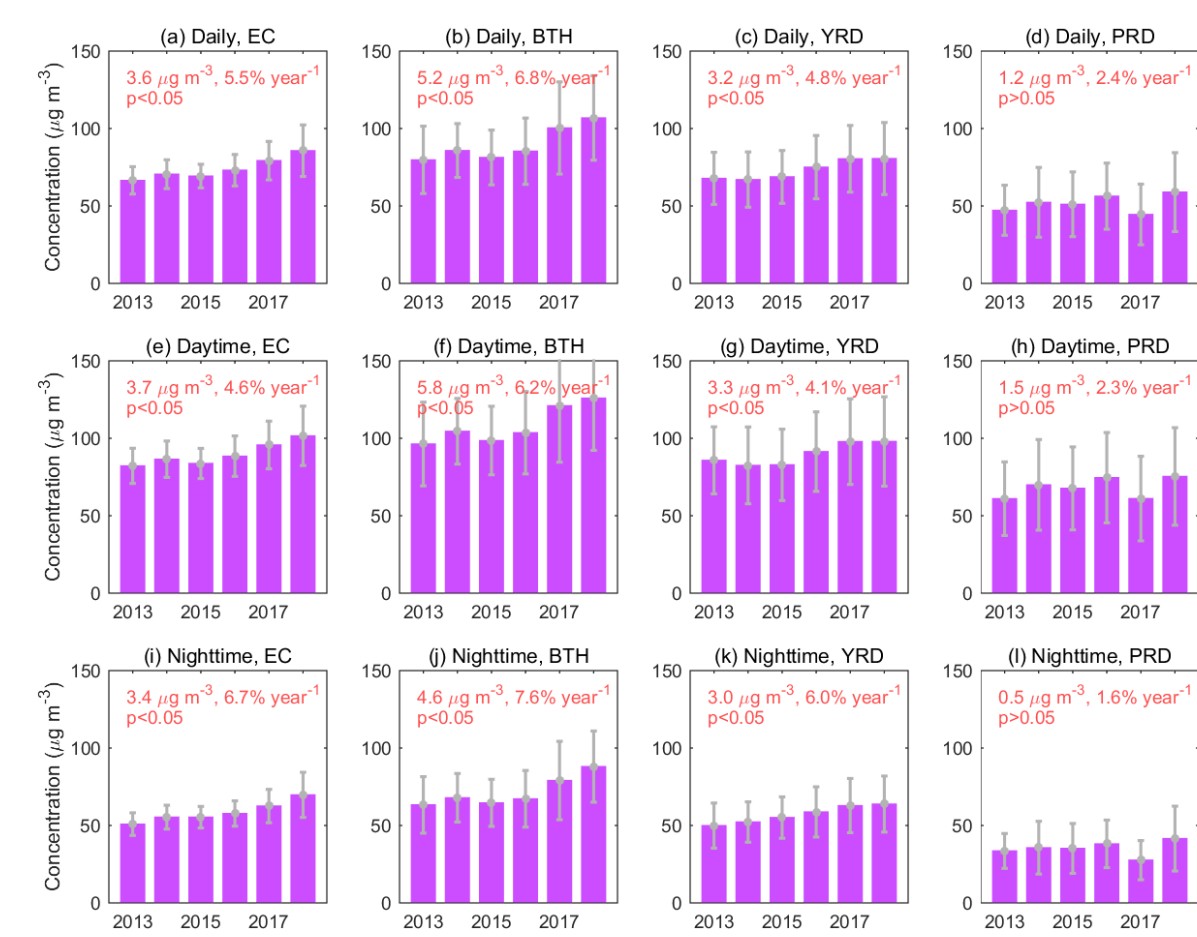

Figure 4. Interannual variations of regional daily mean (1st row), daytime mean (2nd row), and nighttime mean (3rd row) surface ozone concentrations over eastern China and the three subregions in summer from 2013 to 2018. The values were calculated from the observations of the corresponding cities. The error bar indicates two times of the standard deviation. The red numbers are the increasing trends of summer mean ozone in μg m$^{-3}$ and in percentage per year, and the corresponding significant level.



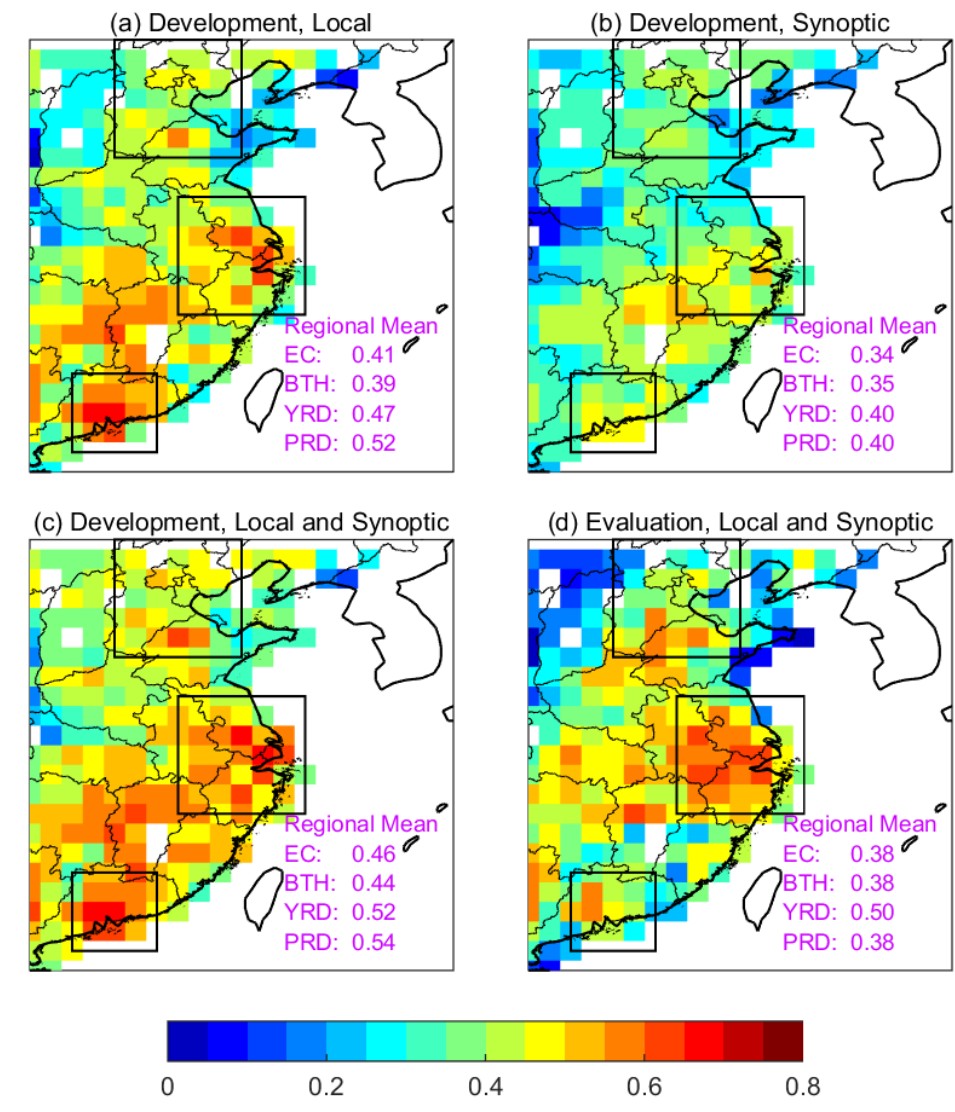

Figure 5. Coefficient of determination ($R^2$) between the observed and predicted daily surface ozone in summer during 2013-2017 when the MLR model is developed (a) with local meteorological variables, (b) with synoptic meteorological factors, and (c) with both of them. (d) The same as (c), but for 2018 when the MLR model is evaluated. The regional mean values are shown in the bottom right corner of each panel. The boxed areas indicate BTH, YRD, and PRD, respectively, in the north, center, and south of the study domain.



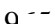

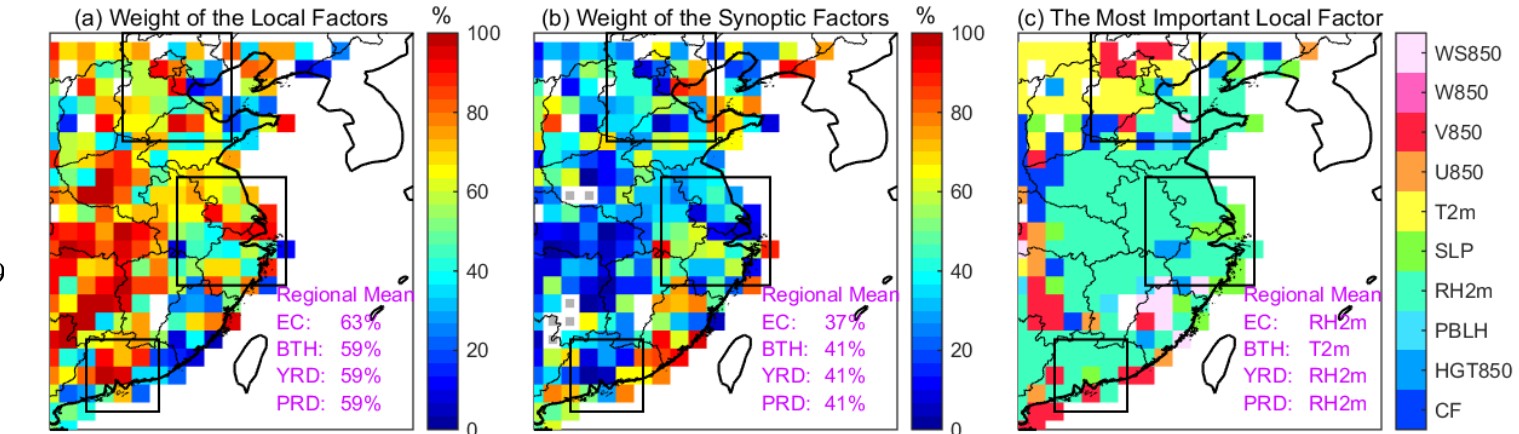

Figure 6. Weights (in %) of (a) local and (b) synoptic meteorological factors in the MLR model describing the daily surface ozone in summer during 2013-2017. (c) The most important variables among the local meteorology for ozone in the MLR model. The regional mean weight in (a) and (b) and meteorological variable in (c) are shown in the bottom right corner of each panel. The grids with grey squares are where the meteorological factors are excluded in the MLR model because they are insignificance ($p > 0.05$). The boxed areas indicate BTH, YRD, and PRD, respectively, in the north, center, and south of the study domain.





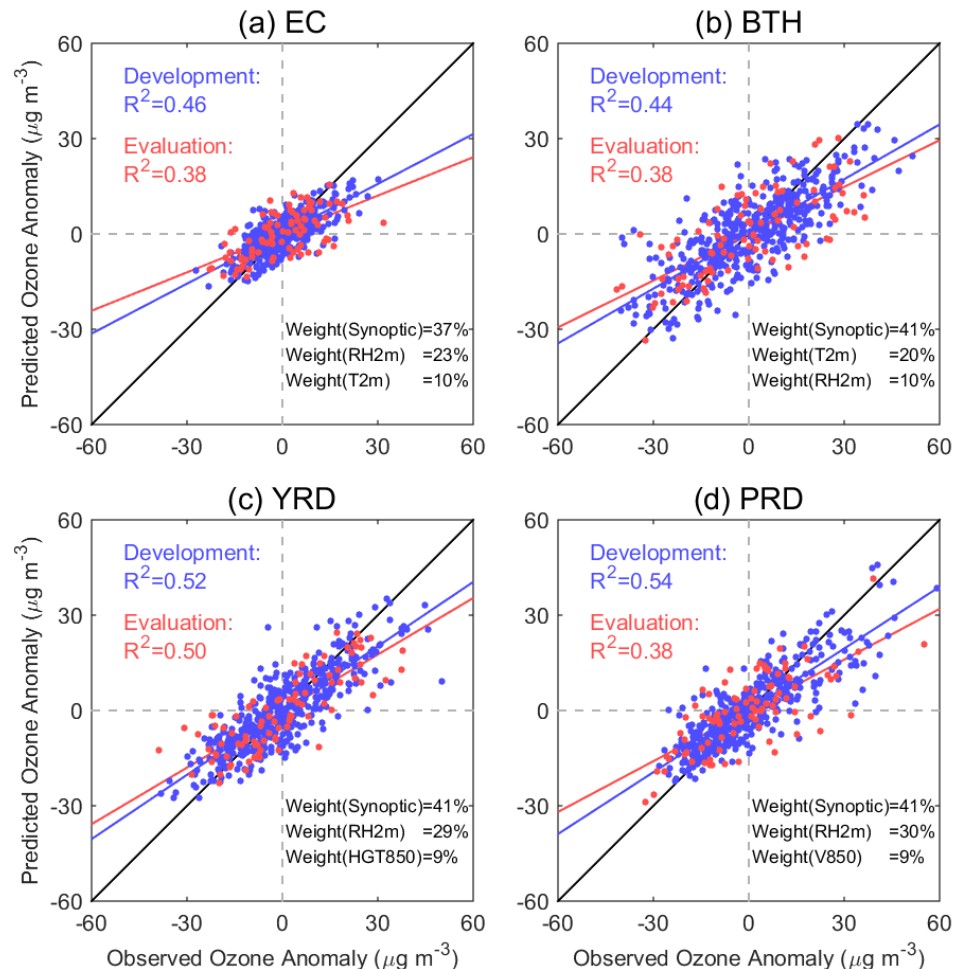

Figure 7. Comparison of daily surface ozone anomalies in summer between the predictions and observations averaged over eastern China and the three subregions. The blue dots show the values in

2013-2017 during which the MLR model is developed, while the red dots are values in 2018 when the model is evaluated. The corresponding linearly fitted lines are in blue and red, respectively, for the model development and evaluation, in comparison with a 1:1 line in black. Weights of the three most dominant meteorological variables in the MLR model for each region are shown in the bottom right corner of each panel.





Figure 8. PSW (P1). The mean (a) Geopotential height at 850 hPa (HGT850), (b) occurrence frequency, and anomalies of (c) observed daily surface ozone, (d) predicted daily surface ozone using MLR, (e) HGT850, (f) relative humidity at 2 m (RH2m), (g) temperature at 2 m (T2m), (h) cloud fraction (CF), (i) planetary boundary layer height (PBLH), (j) meridional wind at 850 hPa (V850), (k) wind speed at 850 hPa (WS850), and (l) air stagnation (AS) in summer during 2013-2018. Anomalies of normalized EASMI and WPSHI (shown in (a)) respectively represent the strength of EASM and WPSH. The red numbers in (b) are the occurrences of PSW in days and in percentage. The purple dots in the boxed area in (a) indicate the cities in BTH, YRD, and PRD, respectively, in the north, center, and south of study domain: eastern China. The boxed areas in (c) and (d) indicate BTH, YRD, and PRD, respectively, in





the north, center, and south of the study domain. The regional mean anomalies of observed and

predicted ozone (±two times of the standard error of the mean) are shown in the bottom right corners of

(c) and (d), respectively.








Figure 9. The same as Figure 8, but for PS (P2).









Figure 10. The same as Figure 8, but for PNECV (P3).









Figure 11. The same as Figure 8, but for PWC (P4).










Figure 12. The same as Figure 8, but for PSWPSH (P5).




Figure 13. The same as Figure 8, but for PTC (P6).





Figure 14. The mean relative anomalies (in %) of observed daily surface ozone under each of the SWPs during 2013-2018. The regional mean anomalies (±two times of the standard error of the mean) are shown in the bottom right corner of each panel. The boxed areas indicate BTH, YRD, and PRD, respectively, in the north, center, and south of the study domain.