# Peer review of "Local and synoptic meteorological influences on daily variability of summertime surface ozone in eastern China"

_Atmospheric Chemistry and Physics, 2019_

## Referee Comment (RC1) · Anonymous Referee #2 · 17 Aug 2019

The authors conducted an extensive analysis of local and synoptic meteorological influences on daily variability in summertime surface ozone in eastern China for the time period of 2013 – 2018. They derived a multiple linear regression (MLR) equation for each grid within the eastern China domain to capture the linear relationships of daily average ozone concentrations as a function of 10 local meteorological variables and 2 synoptic factors, the latter derived using the singular value decomposition (SVD) method. Not to be pedantic, it is an overstatement to call the MLR equation a model. They further examined synoptic weather patterns (SWPs) over eastern China using a self-organizing map (SOM) clustering technique. The MLR and SWPs provides a rich source of information but the authors were short of making a connection between the

two. One interesting point from MLR was, as local meteorological variables, relative humidity in the central and southern parts of eastern China and temperature in the BTH region showing the largest influence on surface ozone concentrations. The study would have been more in-depth should the authors have endeavored to understand the mechanism(s) driving that. Would it be possible to use their SWPs to further understand that point? The authors did use their derived MLR to validate the calculated surface ozone concentrations under the 6 SWPs, but they only showed visual comparisons between the predicted and observed values. It'd make a stronger case if they could show some quantitative comparison.

Most of Section 6 "Discussion and conclusions" repeated the results prior to it with the last paragraph suggesting the potential significance of the study. There was not really much discussion but repetition. I suggest that the section be shortened and changed to "Summary".

Figure 14 is missing from the manuscript.

In the huge body of published work on surface ozone as a pollutant, the majority has used ppbv as units for ozone, and indeed in study of atmospheric trace gases mixing ratios have been used conventionally. The authors' use of mass units was a bit peculiar. I suggest that they provide unit conversion upon the first appearance of the mass units if they insist upon using them.

Some specific comments:

1. Line 36: The first sentence covered both human and vegetation health but the reference cited, Yue et al. (2017), was on vegetation. 2. Lines 78 – 80: Shen et al. (2017a) is not the first and only reference for such a well-established point. There is a huge wealth of research on this point dating back to decades ago. This seems to be a fairly common problem nowadays, that for an extensively, long studied topic, only most recent few studies would be cited whereas a long list of monumental studies leading up to the recent works tend to be left out. In my opinion, we need to do due diligence

to cite the references where credit is due. 3. Line 167: What was "daily surface ozone" meant? Daily average or daily maximum 8-hr average ozone concentrations? Also, the acronym for the latter would be DM8(H)A; it's curious why the authors used "MD8A" instead. 4. Lines 298 – 299: Not clear where this came from. 5. Line 301 – 302: Was "higher meridional wind" enough to bring in "clean and humid marine air to the south" regardless of wind direction? 6. Lines 305-306: How did the authors know that "the impacts of relative humidity on surface ozone are mainly through the chemical processes"? 7. Line 321: why did R2=0.38 qualify to be "strong"? 8. Lines 388: why is precipitation included in the indexes?

---

## Referee Comment (RC2) · Anonymous Referee #1 · 31 Aug 2019

This study explores the local and synoptic meteorological influences on summertime ozone variability in eastern China. The authors have used two different approaches, including MLR and SOM, and their MLR method includes the effects of synoptic patterns. In general, I think this manuscript is well structured and the topic is suitable for ACP. The presentation quality is also very high. So I recommend publication in ACP after fixing only a few minor issues.

**Minor comments.**
L123. Please specify the value of $d_{max}$ used in this study?

L283. It is worthwhile to mention the trends of anthropogenic emissions during this period. During the period of 2013-2017, the NOx emissions have declined (Zheng et al., 2018) and the VOC emissions almost remain constant (Zheng et al., 2018; Shen et al., 2019).

Line 299. The correlation of temperature with ozone is higher in the north than in the south. I guess this is because the temperature gradient along the latitude is much larger in the north than in the south. So the synoptic activities (or jet wind) in the north can be well represented by temperature. But in the south, the marine flux into the continent won't result much changes in temperature but will definitely elevate the relative humidity. As a result, relative humidity displays much stronger correlation with ozone in the south. However, I never found any paper that discusses about the reason of this distinct north-south feature. Maybe the author can simply say like this.

The correlation of temperature with ozone is higher in the north than in the south over eastern China (Figure 2c), which is consistent with the pattern found in the US (Camalier et al., 2007; Shen et al., 2016).

Line 305. Relative humidity can also be a strong indicator of transport in the southern China. So I won't use the words "mainly" here.

Figure 5. I would strongly recommend using the leave-one-out cross validation to avoid overfitting if the authors haven't done this.

Figure 6. I don't know if there is a way to separate the effects of local meteorology and synoptic patterns. They are so closely related. I would suggest deleting this figure and also the text related.

Zheng, B., et al.: Trends in China's anthropogenic emissions since 2010 as the consequence of clean air actions, Atmos. Chem. Phys., 18, 14095–14111, https://doi.org/10.5194/acp-18-14095- 2018, 2018.

Shen, L., L. J. Mickley and E. Gilleland, Impact of increasing heatwaves on U.S. ozone episodes in the 2050s: Results from a multi-model analysis using extreme value theory, Geophys. Res. Lett., 43, doi:10.1002/2016GL068432, 2016.

Camalier, L., Cox, W., and Dolwick, P.: The effects of meteorology on ozone in urban areas and their use in assessing ozone trends, Atmos. Environ., 41, 7127-7137, https://doi.org/10.1016/j.atmosenv.2007.04.061, 2007.

Shen, L., et al., 2005-2016 trends of formaldehyde columns over China observed by satellites: increasing anthropogenic emissions of volatile organic compounds and decreasing agricultural fire emissions, Geophys. Res. Lett., 46, 4468-4475, 2019

---

## Author Comment (AC1) · 10 Oct 2019

*We thank the reviewers for their valuable comments and suggestions to improve our manuscript. We have made revisions accordingly. The point-to-point responses are provided below in Italic. The comparison of our manuscript between this version and the previous version is also provided.*

Anonymous Referee #2

The authors conducted an extensive analysis of local and synoptic meteorological influences on daily variability in summertime surface ozone in eastern China for the time period of 2013-2018. They derived a multiple linear regression (MLR) equation for each grid within the eastern China domain to capture the linear relationships of daily average ozone concentrations as a function of 10 local meteorological variables and 2 synoptic factors, the latter derived using the singular value decomposition (SVD) method. Not to be pedantic, it is an overstatement to call the MLR equation a model. They further examined synoptic weather patterns (SWPs) over eastern China using a self-organizing map (SOM) clustering technique. The MLR and SWPs provides a rich source of information but the authors were short of making a connection between the two. One interesting point from MLR was, as local meteorological variables, relative humidity in the central and southern parts of eastern China and temperature in the BTH region showing the largest influence on surface ozone concentrations. The study would have been more in-depth should the authors have endeavored to understand the mechanism(s) driving that. Would it be possible to use their SWPs to further understand that point? The authors did use their derived MLR to validate the calculated surface ozone concentrations under the 6 SWPs, but they only showed visual comparisons between the predicted and observed values. It'd make a stronger case if they could show some quantitative comparison. Most of Section 6 "Discussion and conclusions" repeated the results prior to it with the last paragraph suggesting the potential significance of the study. There was not really much discussion but repetition. I suggest that the section be shortened and changed to

"Summary". Figure 14 is missing from the manuscript. In the huge body of published
work on surface ozone as a pollutant, the majority has used ppbv as units for ozone,
and indeed in study of atmospheric trace gases mixing ratios have been used
conventionally. The authors' use of mass units was a bit peculiar. I suggest that they
provide unit conversion upon the first appearance of the mass units if they insist upon
using them.

*Thanks for the valuable comments. The following is our answers to the reviewer's
questions.*

*By fitting a linear equation, the MLR predict the response variable by using several
explanatory variables. As a simple and basic regression model, it has been widely
used in the prediction of atmospheric pollutants (Kutner et al., 2004; Gao et al.,
2019; Li et al., 2019). In this study, the MLR is applied to predict surface ozone in
eastern China with the predictors of meteorological factors. In this revision, we
further used the leave-one-out cross validation (Section 2.2, Lines 204-208) to avoid
overfitting of the MLR. The MLR shows strong performance with a regional mean
coefficient of determination ($R^2$) of 43% (Figure 5a).*

*In this revision, we combined the MLR and SOM to reveal the most important local
meteorological factor for ozone variability under each of the six SWPs (Figure S6).
The MLR was conducted under each of the SWPs with the same procedures for the
full summer. The most important meteorological variable for ozone over some areas
in eastern China may vary with the prevailing SWP (Figure S6). The dominant driver
in PRD is meridional wind at 850 hPa under PSW (P1), PS (P2), and PSWPSH (P5),
demonstrating the significant influences of marine air inflow. Controlled by the
typhoon system, the most important factor over some coastal areas is zonal wind at
hPa under PTC (P6). The analysis has been added in Lines 486-492.*

*Results from the MLR show that among local meteorological factors, relative humidity is the foremost influential variable for summertime surface ozone over most locations in the center and south of eastern China including YRD and PRD, while*

*temperature is more important in the north including BTH. Such a difference between the north and south were also found in the eastern United States by previous studies (Camalier et al., 2007; Porter et al., 2015). The difference is possibly related with both ozone photochemistry and synoptic influences. The relative importance of relative humidity and temperature to ozone photochemistry may vary with location,*

*because of different atmospheric environments. Moreover, the sensitivities of relative humidity and temperature to synoptic systems may change with the location as well. However, until now, there are no strong evidence to explain it.*

*We have added some statistical comparisons between the observations and*

*predictions of averaged ozone anomalies under each of the SWPs in Table S1. The mean absolute error (MAE) ranges 1.0-2.2 $\mu g\ m^{-3}$ and the root mean square error (RMSE) ranges 1.4-2.8 $\mu g\ m^{-3}$.*

*We have renamed Section 6 as 'Summary' and shortened this section.*

*In the last version, Figure 14 showed the relative anomalies of observed surface ozone under the six SWPs, giving an additional explanation of Figures 8d, 9d, 10d, 11d, 12d, and 13d. In this revision, Figure 14 is moved into the supplement as Figure S5. We reserve the regional mean relative anomalies in Table 1.*

*We used the unit '$\mu g\ m^{-3}$' to keep consistency with that for China national air quality standard. For ozone, 1 $\mu g\ m^{-3}$ equals to 0.47 ppbv at 273 K and 1013.25 hPa. The unit conversion is added in Lines 116-117.*

*Kutner, M.H., Nachtsheim, C.J., Neter, J., Li, W., 2004. Applied Linear Statistical*
          *Models. McGraw-Hill/Irwin, New York, NY, USA.*

      *Gao, M., Sherman, P., Song, S., Yu, Y., Wu, Z., and McElroy, M. B.: Seasonal*
          *prediction of Indian wintertime aerosol pollution using the ocean memory effect,*
          *Science Advances, 5, eaav4157, 10.1126/sciadv.aav4157, 2019.*

*Li, K., Jacob, D. J., Liao, H., Shen, L., Zhang, Q., and Bates, K. H.: Anthropogenic*
          *drivers of 2013-2017 trends in summer surface ozone in China, Proc. Natl. Acad.*
          *Sci. U. S. A., 116, 422, https://doi.org/10.1073/pnas.1812168116, 2019.*

Some specific comments:

1. Line 36: The first sentence covered both human and vegetation health but the
      reference cited, Yue et al. (2017), was on vegetation.

      *Thanks. A reference for human health is added (Jerrett et al., 2009).*

      *Jerrett, M., Burnett, R. T., Pope, C. A., Ito, K., Thurston, G., Krewski, D., Shi, Y.,*
*Calle, E., and Thun, M.: Long-term ozone exposure and mortality, N. Engl. J. Med.,*
          *360, 1085-1095, https://doi.org/10.1056/NEJMoa0803894, 2009.*

      2. Lines 78-80: Shen et al. (2017a) is not the first and only reference for such a well-
      established point. There is a huge wealth of research on this point dating back to
decades ago. This seems to be a fairly common problem nowadays, that for an
      extensively, long studied topic, only most recent few studies would be cited whereas a
      long list of monumental studies leading up to the recent works tend to be left out. In
      my opinion, we need to do due diligence to cite the references where credit is due.

      *Thanks for the points. Several reliable studies are added (Bloomfield e al., 1996;*
*Davis et al., 1998; Zanis et al., 2000, 2011; Ordóñez et al., 2005; Camalier et al.,*
      *2007).*

*Bloomfield, P., Royle, J. A., Steinberg, L. J., and Yang, Q.: Accounting for meteorological effects in measuring urban ozone levels and trends, Atmos.*

*Environ., 30, 3067-3077, https://doi.org/10.1016/1352-2310(95)00347-9, 1996.*

*Camalier, L., Cox, W., and Dolwick, P.: The effects of meteorology on ozone in urban areas and their use in assessing ozone trends, Atmos. Environ., 41, 7127-7137, https://doi.org/10.1016/j.atmosenv.2007.04.061, 2007.*

*Davis, J. M., Eder, B. K., Nychka, D., and Yang, Q.: Modeling the effects of*

*meteorology on ozone in Houston using cluster analysis and generalized additive models, Atmos. Environ., 32, 2505-2520, https://doi.org/10.1016/S1352-2310(98)00008-9, 1998.*

*Ordóñez, C., Mathis, H., Furger, M., Henne, S., Hüglin, C., Staehelin, J., and Prévôt, A. S. H.: Changes of daily surface ozone maxima in Switzerland in all seasons from*

*1992 to 2002 and discussion of summer 2003, Atmos. Chem. Phys., 5, 1187-1203, https://doi.org/10.5194/acp-5-1187-2005, 2005.*

*Zanis, P., Monks, P. S., Schuepbach, E., Carpenter, L. J., Green, T. J., Mills, G. P., Bauguitte, S., and Penkett, S. A.: In situ ozone production under free tropospheric conditions during FREETEX '98 in the Swiss Alps, J. Geophys. Res.-Atmos., 105,*

*24223-24234, https://doi.org/10.1029/2000JD900229, 2000.*

*Zanis, P., Katragkou, E., Tegoulias, I., Poupkou, A., Melas, D., Huszar, P., and Giorgi, F.: Evaluation of near surface ozone in air quality simulations forced by a regional climate model over Europe for the period 1991-2000, Atmos. Environ., 45, 6489-6500, https://doi.org/10.1016/j.atmosenv.2011.09.001, 2011.*

3. Line 167: What was "daily surface ozone" meant? Daily average or daily maximum 8-hr average ozone concentrations? Also, the acronym for the latter would be DM8(H)A; it's curious why the authors used "MD8A" instead.

*Thanks. In this study, the ozone-weather relationship is examined using the daily*

*mean ozone and meteorological data. We have clarified this in this revision (Lines*

*122-123).*

*'Daily maximum 8-hour average' can also be stated as 'maximum daily 8 h mean' (Silver et al., 2018), 'daily maximum 8-hour running mean' (Fleming et al., 2018), or*
*'maximum daily average 8-h' (Lefohn et al., 2018). Conventionally, these expressions can all be termed as 'MDA8'.*

*Silver, B., Reddington, C. L., Arnold, S. R., and Spracklen, D. V.: Substantial changes in air pollution across China during 2015-2017, Environ. Res. Lett., 13, 114012,*
*https://doi.org/10.1088/1748-9326/aae718, 2018.*
*Fleming, ZL, et al. 2018 Tropospheric Ozone Assessment Report: Present-day ozone distribution and trends relevant to human health. Elem Sci Anth, 6: 12. https://doi.org/10.1525/elementa.273.*
*Lefohn, AS, et al. 2018 Tropospheric ozone assessment report: Global ozone metrics*
*for climate change, human health, and crop/ecosystem research. Elem Sci Anth, 6: 28. https://doi.org/10.1525/elementa.279.*

4. Lines 298-299: Not clear where this came from.

*Thanks. The sentence has been revised and the unclear statement has been removed.*

5. Line 301-302: Was "higher meridional wind" enough to bring in "clean and humid marine air to the south" regardless of wind direction?

*Thanks. In summer, the south-westerly monsoon wind prevails over eastern China (Figure S3). In most of the days in summer, the meridional wind blows from the south*
*to the north. We added the explanation in Lines 320-321 in this revision.*

6. Lines 305-306: How did the authors know that "the impacts of relative humidity on surface ozone are mainly through the chemical processes"?

*The expressions have been revised (Lines 310-316). Relative humidity can influence ozone through various processes. Atmospheric water vapor can directly influence ozone concentrations by $HO_x$ ($HO_x$=OH+H+peroxy radicals) chemistry with complicated regimes (Lu et al., 2019b). Moreover, a higher relative humidity is usually associated with more fractions of clouds, which can slow the photochemical production of surface ozone. In addition, higher relative humidity may somewhat be linked with larger atmospheric instability, favoring the dispersion of surface ozone (Camalier et al., 2007).*

7. Line 321: why did R2=0.38 qualify to be "strong"?

*The sentences related to $R^2$ have been revised. In this revision, the leave-one-out cross validation is used to avoid overfitting of the MLR. The regional mean cross-validated $R^2$ over eastern China is 43%, indicating strong performance of the MLR. Because of the large sample size (552 samples), the statistical results are at a very high significant level, with p value being far below 0.01.*

8. Lines 388: why is precipitation included in the indexes?

*The air stagnation index used in this study is a common index to assess air mass stagnation (Wang and Angell, 1999; Horton et al., 2012). Precipitation is often accompanied with deep or shallow convection. So, a day is considered to meet stagnation criteria, when daily total precipitation is less than 1 mm, which means a dry day.*

*Wang, J.X.L., and J.K. Angell, 1999: Air Stagnation Climatology for the United States (1948-1998). NOAA/Air Resources Laboratory ATLAS, No.1.*

[revised manuscript text omitted]

---

## Author Comment (AC2) · 10 Oct 2019

*We thank the reviewers for their valuable comments and suggestions to improve our manuscript. We have made revisions accordingly. The point-to-point responses are provided below in Italic. The comparison of our manuscript between this version and the previous version is also provided.*

Anonymous Referee #1:

This study explores the local and synoptic meteorological influences on summertime ozone variability in eastern China. The authors have used two different approaches, including MLR and SOM, and their MLR method includes the effects of synoptic

10 patterns. In general, I think this manuscript is well structured and the topic is suitable for ACP. The presentation quality is also very high. So I recommend publication in ACP after fixing only a few minor issues.

*The reviewer's encouragement is much appreciated.*

15 Minor comments.

L123. Please specify the value of dmax used in this study?

*'$d_{max}$' is specified as 1-degree distance in latitude-longitude grid. Please see Lines 132-133.*

20 L283. It is worthwhile to mention the trends of anthropogenic emissions during this period. During the period of 2013-2017, the NOx emissions have declined (Zheng et al., 2018) and the VOC emissions almost remain constant (Zheng et al., 2018; Shen et al., 2019).

*Thanks. Added. See Lines 293-295.*

Line 299. The correlation of temperature with ozone is higher in the north than in the south. I guess this is because the temperature gradient along the latitude is much larger in the north than in the south. So the synoptic activities (or jet wind) in the

north can be well represented by temperature. But in the south, the marine flux into the continent won't result much changes in temperature but will definitely elevate the relative humidity. As a result, relative humidity displays much stronger correlation with ozone in the south. However, I never found any paper that discusses about the reason of this distinct north-south feature. Maybe the author can simply say like this.

The correlation of temperature with ozone is higher in the north than in the south over eastern China (Figure 2c), which is consistent with the pattern found in the US (Camalier et al., 2007; Shen et al., 2016).

*Thanks. Revised following the reviewer's suggestions. Please see Lines 317-319.*

Line 305. Relative humidity can also be a strong indicator of transport in the southern China. So I won't use the words "mainly" here.

*Thanks. Revised. Please see Lines 310-316.*

Figure 5. I would strongly recommend using the leave-one-out cross validation to avoid overfitting if the authors haven't done this.

*Thanks. We have used the leave-one-out cross validation in this revision. Please see Lines 204-208.*

Figure 6. I don't know if there is a way to separate the effects of local meteorology and synoptic patterns. They are so closely related. I would suggest deleting this figure and also the text related.

*Thanks for the suggestion. We have deleted the comparison of the weight between local meteorology and synoptic patterns and reserved the comparison between different local meteorological variables in this revision.*

[revised manuscript text omitted]

regional mean v̶a̶l̶u̶e̶s̶ ̶o̶fozone by 5.2 μg m$^{-3}$ (7.5%) over YRD and 6.7 μg m$^{-3}$ (11.8%)

over PRD (F̶i̶g̶u̶r̶e̶s̶ ̶1̶1̶c̶Figure 10c and 1̶4̶dTable 1). Mean negative ozone anomalies

of -4.8 μg m$^{-3}$ (-5.1%) are observed over BTH in PWC (Figure 10c and Table 1).

620

PSW̶PSH (P5) occurs in late summer (Figure 1̶2̶a̶11b), when Meiyu breaks in the

Yangtze River and the rain belt j̶u̶m̶p̶sshifts to North China (Ding and Chan, 2005). In

PSWPSH, the WPSH is the strongest and extends westward t̶h̶e̶ mostly (Figure

1̶2̶a̶11a). Thus, relative humidity is lower than the seasonal mean over YRD and

625 higher than the seasonal mean over BTH (Figure 1̶2̶f̶11f). Meantime, stable weather

conditions occur more frequently over YRD (Figure 1̶2̶l̶11l). Therefore, ozone

accumulates over YRD in PSWPSH with a regional mean enhancement of 1.8 μg m$^{-3}$

(2.5%) (F̶i̶g̶u̶r̶e̶s̶ ̶1̶2̶c̶Figure 11c and 1̶4̶eTable 1). Surface ozone decreases by 0.8

(1.4%) and 5.0 μg m$^{-3}$ (8.9%̶)%), respectively,̶ over BTH and PRD under this SWP

630 (F̶i̶g̶u̶r̶e̶s̶ ̶1̶2̶c̶Figure 11c and 1̶4̶eTable 1).

[revised manuscript text omitted]

---

## Author Response (AR2)

*Dear Prof. Tong Zhu,*

*We thank the reviewer for the valuable comments that help us improve our manuscript. We have made a revision accordingly. The point-to-point responses are provided below in Italic. The comparison of our manuscript between this version and the previous version is also provided. Please let me know should you have future question.*

*Thanks and best regards,*

*Jane Liu on behalf of all the co-authors*

Anonymous Referee #2:

The authors addressed most of my comments. In particular the addition of Figure S6, in my opinion, is a key finding, where the most important meteorological factor was identified for each study domain. The reason why there was regionality in those factors is quite complex, which warrants future investigation. I would use this figure in the main text. Other than that, I have a couple of minor suggestions. One is the use of the term "trend" for the increases discussed in lines 287-295. I'd say only Shen et al. (2019a)'s data record was long enough to qualify the use of "trend" whereas the remaining studies had merely a couple of years of data. I would be using "trend" sparingly in those other cases. The other is the use of singular and plural nouns as well as articles. Overall this is a very solid study. I congratulate the authors for their nice contribution to the community.

*Thanks for the comments and suggestions. Figure S6 has been moved to the main text as Figure 13. We have checked the use of "trend" and made revisions accordingly. We have also checked the use of singular and plural nouns in the manuscript. Please see the tracking change for these revisions.*

[revised manuscript text omitted]